# Comparative Analysis of Reinforced Asphalt Concrete Overlays: Effects of Thickness and Temperature

**DOI:** 10.3390/ma16175990

**Published:** 2023-08-31

**Authors:** Amjad H. Albayati, Yasmin S. Ajool, Abbas A. Allawi

**Affiliations:** Department of Civil Engineering, University of Baghdad, Baghdad 10071, Iraq; yasmeen.dawood2001m@coeng.uobaghdad.edu.iq (Y.S.A.); a.allawi@uobaghdad.edu.iq (A.A.A.)

**Keywords:** reflection cracking, overlay, geotextile, Kevlar fiber, chopped fiber, fracture energy

## Abstract

Reflection cracking in asphalt concrete (AC) overlays is a common form of pavement deterioration that occurs when underlying cracks and joints in the pavement structure propagate through an overlay due to thermal and traffic-induced movement, ultimately degrading the pavement’s lifespan and performance. This study aims to determine how alterations in overlay thickness and temperature conditions, the incorporation of chopped fibers, and the use of geotextiles influence the overlay’s capacity to postpone the occurrence of reflection cracking. To achieve the above objective, a total of 36 prism specimens were prepared and tested using an overlay testing machine (OTM). The variables considered in this study were the thickness of the overlay (40, 50, and 60 mm), temperature (20, 30, and 40 °C), mix type (reference mix and mix modified with 10% chopped fibers by weight of asphalt cement), and the inclusion of geotextile fabric at two positions (one-third of the depth from the base and at the bottom). The research outcomes revealed that a decreased temperature and thicker overlay led to a higher resistance to crack initiation and full propagation, as indicated by the values of critical fracture energy (*Gc*) and crack progression rate (CPR). Furthermore, the study observed the enhanced crack resistance of overlays in the presence of geotextiles, whether at the bottom or one-third of the depth from the bottom, with superior performance of the former. Despite a slight enhancement in certain properties, the incorporation of chopped fibers in the overlays did not substantially improve the overall performance compared to the reference specimens. Overall, the study provides valuable insights into the variables that influence the ability of AC overlays to mitigate reflection cracking. These findings will aid engineers and designers in making informed decisions regarding overlay design and construction.

## 1. Introduction

Asphalt pavements are typically designed to last 20 years. This period is considered their design lifespan [1,2], during which the roads are used to their fullest potential with a comfortable riding quality, assuming optimal maintenance. However, a rise in traffic volume, the influence of changing weather conditions, and improper or delayed maintenance can cause road surfaces to deteriorate before their expected lifespan [3]. One of the key strategies for rehabilitating the existing hot-mix asphalt (HMA) and Portland cement concrete (PCC) pavements is an asphalt concrete overlay. Nevertheless, experience has proven that existing cracks on the old road surface tend to rapidly extend into the new overlay due to variable weather and traffic conditions [4]. This is known as “reflection cracking,” which occurs when the new surface replicates the same cracking pattern as the old one beneath it.

Reflection cracking in asphalt concrete (AC) overlays is a significant issue that researchers have addressed extensively in the literature due to its potential to reduce pavement serviceability and performance. There is a consensus that reflection cracking is generally caused by thermal and traffic-induced movements, as shown in Figure 1. The crack growth is induced by bending or shearing from passing traffic loads or by temperature changes. 

The ongoing challenge of reflection cracking in asphalt overlays has prompted the development of various laboratory testing methods that have played a pivotal role in understanding this phenomenon. Wheel-tracking tests, such as the Hamburg wheel-tracking test (HWTT) and the loaded wheel tester (LWT), have been used extensively to simulate traffic loading conditions and the resulting reflection cracking. However, as noted by Karlsson and Isacsson [5], these tests are best suited for rutting analysis and may not accurately capture the specific distress mechanisms involved in reflection cracking. Additionally, bending beam tests such as the four-point bending (4PB) test and the three-point bending (3PB) test have been utilized to evaluate the fatigue cracking behavior of asphalt mixes. These tests are generally performed under controlled strain or stress conditions to evaluate the fatigue life of the material. Under the right conditions, they can provide valuable insights into reflection cracking behavior [6]. On the other hand, direct tension tests such as the indirect tensile test (IDT) and the semi-circular bending (SCB) test are commonly used to evaluate the tensile strength and fracture resistance of asphalt mixes. These properties are vital for understanding the mix’s susceptibility to reflection cracking [7]. A significant shift occurred in the field of laboratory tests when researchers began using the Texas overlay test (TOT), which simulates reflection cracking by applying repetitive strain to asphalt concrete overlay specimens. The TOT provides key measurements such as the number of load cycles to failure (Nf) and the maximum tensile load, which are instrumental in understanding and mitigating reflection cracking [8]. Garcia et al. [9] introduced two alternative parameters, the critical fracture energy (*Gc*) and the crack progression rate (CPR), to assess the performance of AC specimens during the OT’s crack initiation and propagation stages. The reliability and uniformity of these parameters were examined based on 60 OT test outcomes using an identical mix. This assessment was part of a collaborative study between the Texas Department of Transportation (TxDOT) and The University of Texas at El Paso (UTEP). The results from both laboratories indicated consistent and commendable repeatability for the parameters proposed by Garcia et al. Despite the success of some tests in characterizing the overlay reflection cracking, the OT test is more popular since the other tests have been criticized for the variability of their results, which can be highly affected by factors such as specimen preparation and testing temperature [10]. Therefore, in this research, the OT method was adopted to characterize the overlay reflection cracking, and four parameters were considered to evaluate the specimen’s ability to withstand the initiation as well as propagation of reflection cracking; those parameters were *Gc*, CPR, initial tensile load and Nf. 

In recent years, various research efforts have been directed towards elucidating the factors contributing to reflection cracking and developing mitigation strategies, as shown in Table 1. It can be seen that a significant amount of work has been investigated regarding individual strategies in each study to minimize reflection cracking. Additionally, the mitigation strategies proposed are not universally effective, and their performance often varies based on local conditions, suggesting a need for adaptable and region-specific solutions. A notable gap that can be recorded in reviewing the existing literature is the absence of the synergic effect of temperature and overlay thicknesses on reflection cracking mitigation strategies. Temperature plays a crucial role in the performance of the overlay due to the viscoelastic properties of asphalt concrete overlays. On the other hand, the thickness of the overlay is intrinsically linked to its ability to absorb stress and distribute loads. To bridge the above gap, this research aimed to delve into the impact of different mitigation strategies under the combined effects of variable temperatures and overlay thicknesses. The results of this work could guide future research and practice toward more effective, sustainable, and resilient asphalt concrete overlays.

## 2. Materials and Testing Methods

This paragraph outlines the materials used in this study and the testing program designed to assess the impact of various variables on the ability of asphalt concrete overlays to mitigate reflection cracking.

### 2.1. Materials

#### 2.1.1. Asphalt Cement

The asphalt cement utilized for this study was sourced from the Doura oil refinery located in the southwest region of Baghdad. It tested in line with the Superpave performance grade criteria. Table 2 presents the test results, which confirmed that the asphalt cement complied with the specifications of the PG 64-16 grade.

#### 2.1.2. Aggregate

For this experimental endeavor, the chosen aggregate was crushed quartz brought from the Amanat Baghdad asphalt concrete mix plant located north of Baghdad. The coarse and fine components of this aggregate were separated using sieves and then recombined in appropriate proportions to meet the grading requirements for a Type III A mix. This mix has a nominal maximum size of 12.5 mm (0.5 inch) and is commonly utilized for overlay construction in accordance with SCRB/R9 [23] specifications. The gradation curve of the aggregate is depicted in Figure 2.

Routine assessments were performed on the aggregate to establish its physical characteristics. Table 3 presents the results of these tests, along with the SCRB-prescribed specification boundaries. The outcomes of the tests verified that the selected aggregate satisfied the SCRB requirements. 

#### 2.1.3. Mineral Filler

The mineral filler employed was a non-plastic substance that was passed through a No. 200 sieve (0.075 mm). This filler was ordinary Portland cement sourced from a cement factory in Kubaissa (west of Iraq). The physical properties of this filler are detailed in Table 4.

#### 2.1.4. Chopped Fiber

The chopped glass fiber type was used with a dosage of 10% by weight of asphalt cement as a modifier for asphalt concrete mixes of overlays in this research. The physical properties of the fibers are listed in Table 5. A photograph of the chopped fibers is shown in Figure 3.

#### 2.1.5. Geotextile

Kevlar Aramid fiber was used as the geotextile to reinforce the asphalt concrete in this research. Its photograph and physical properties are exhibited in Figure 4 and Table 6, respectively. The geotextile was trimmed into strips of 400 × 300 mm, the same as the dimensions of the slab specimen. It was placed at two locations: the bottom of the specimen and one-third of the depth from the bottom. To ensure better adhesion of the geotextile to asphalt concrete and to keep the asphalt cement constant in the mixture, the geotextile strip was coated with liquid asphalt (tack coat) before placing it in the specimen. 

### 2.2. Mix Preparation and Specimen Legend

Using dry sieve analysis, the aggregate was sieved and sorted into fractions as retained on each of the following sieves: 19, 12.5, 9.5, 4.75, 2.36, 0.3, 0.075 mm, and a pan. The aggregate gradation curve exhibited in Figure 1 represents the mid-range of gradation for asphalt concrete mix Type III A as defined by SCRB/R9 [23]. The aggregate was blended in the specimen preparation bowl to the necessary weight depending on the specimen geometry and test type. The cylindrical specimen of the Marshall test was 1150 kg, whereas the slab specimen, which was then trimmed to prisms for the OT test, was 10,671, 13,340, and 16,000 kg for the slabs with 40, 50, and 60 mm depths. The blend was homogeneously mixed for two minutes and heated to 150 °C for 2 hrs. in a temperature-controlled oven. The bowl was then balanced, and the calculated amount of asphalt cement, which had also been heated to a temperature of 150–155 °C (corresponding to a binder viscosity of 170 ± 20 c.St), was poured into the mixing bowl. 

On the hot plate, the contents of the bowl were thoroughly mixed for two minutes. In the case of the specimen preparation with the chopped fibers, the mixing time was extended by three minutes, and the chopped fibers were introduced into the mixture uniformly. To ensure uniform compaction temperature, the bowl and its contents were placed in the oven for 10 min at 140 °C. During this time, the compaction mold, which had been preheated to 100 °C, was prepared. Following that, the batched material was placed in the mold and compacted according to the test type’s requirements. The Marshall compactor was used in the case of the cylindrical specimen preparation, whereas for the slab specimen, the roller compactor was implemented. The process of compacting in the roller compactor was carried out in a sequence of five steps, ranging from 0.5 kN to 4 kN of force. Different loading cycles were applied for each of these steps. The compaction procedure ended once the density of the sample matched the Marshall density. When utilizing the geotextile at the depth of one-third from the bottom, the compaction occurred in two stages. Initially, one-third of the mixture weight was placed into the mold, and the first stage of compaction was achieved using 0.5 kN of force over 5 loading cycles. Subsequently, the geotextile was positioned on the upper surface of the compacted specimen. Following this, the remaining mixture weight was added to the mold, and the process of compaction was continued. Some photographs presented in Figure 5 show the compaction of mixtures as well as specimens. The slab specimen with dimensions of 300 mm width and 400 mm length was trimmed using a rock cutter to obtain a prism with dimensions of 375 mm length and 75 mm width. Each slab specimen was trimmed into four prisms; Table 7 lists the nomenclature and description of each mix type for the convenience of reference during the testing portion of this research.

### 2.3. Testing Methods

Marshall test

The measurement of resistance to plastic flow was undertaken using the Marshall method, as stipulated by ASTM (D6927) [24]. Marshall stability refers to the peak load a specimen can withstand prior to failure, while Marshall flow describes the total vertical plastic deformation of the specimen. In addition, the specimen’s volumetric characteristics, such as the percentage of air voids (%AV), the void content in mineral aggregate (%VMA), and the voids filled with asphalt (%VFA), were determined based on the specimen’s bulk specific gravity (ASTM-D2726) [25] and the theoretical specific gravity of the mixtures (ASTM-D2041) [26].

The specimen preparation for this test included compaction on both sides with 75 blows each; this level of compaction simulates pavement exposure to high traffic volumes (>10^6^ ESAL). Subsequently, the specimens were removed from their molds and submerged in water for a duration of 30 to 45 min in preparation for the Marshall tests. Each test was performed on three samples, and the results were averaged.

Overlay test

The overlay test (OT) is employed to assess the susceptibility of asphalt mixtures to reflection cracking. The test was conducted on asphalt beams measuring 375 mm in length, 75 mm in width, and varying depths of 40, 50, and 60 mm. To best simulate the field condition, the asphalt concrete specimen was bonded with epoxy glue to two prisms of cement concrete (PCC), each one with a half dimension of the AC prism and 75 mm height; these prisms formed a base for the AC specimen and represented the concrete pavement over which the asphalt concrete overlay was placed in the field. Thereafter, the AC specimen with PCC was bonded to the horizontal platen that existed in the overlay tester machine. The two horizontal platens, as well as the two PCCs, were separated with a gap of 5 mm, which represents the width of joints or cracks in overlayed pavement. Photographs presented in Figure 6 show the specimens and test setup. The test was carried out in a displacement-controlled mode, applying one loading cycle every 10 s as per the proposed ASTM WK26816 protocol. The sliding platen moved in a cyclical triangular pattern, achieving a consistent maximum displacement of 0.635 mm (0.025 inches) at testing temperatures of 20˚C, 30 ˚C, and 40 ˚C. The main result derived from the OT test was the number of load cycles applied to failure (Nf). As per the definition by Zhou and Scullion [27], a specimen is considered to have failed when its maximum load at a particular cycle decreases to 93% below the highest load from the initial cycle. Additionally, the typical parameters of interest in this test were the critical fracture energy (*Gc*) and crack progression rate (CPR). 

The cycles of the applied displacement, load, and corresponding time histories are shown in Figure 7. Additionally, the change in peak tensile load with respect to the number of cycles and the typical first hysteresis loop is shown in the figure. The *Gc* of the specimen is calculated as per the following equation.
$$Gc=\frac{W}{A}$$
where *W* is the area under the curve bounded by the zero-load axis and the vertical axis of the maximum tension load, and *A* is the cross-sectional area of the specimen (width multiplied by height). The *Gc* quantifies the energy required to initiate a crack on the bottom of the specimen in the first loading cycle. The higher the value is, the more energy the AC mix will require to initiate the crack, i.e., greater resistance to reflection cracking imitation. On the other hand, the CPR is the power term obtained from the fitting of a power curve to the load reduction curve and is used for characterizing the resistance to cracking during the propagation phase. The greater the absolute value of the crack progression rate, the faster the crack propagates through the AC specimen and the shorter the AC life will be.

## 3. Mix Design

The Marshall method based on ASTM (D6927) [24] was used to accomplish a comprehensive mix design for specimens prepared by applying 75 blows to each side using an automatic Marshall compactor. According to AI’s manual series No. 2 [28], the optimal asphalt content (OAC) is derived by averaging the three asphalt content values that produce the peak stability, maximum density, and 4 percent air void content. Five percentages of asphalt cement were implemented, starting from 4% by weight of the total mix with an increment rate of 0.5 percent. These were 4.0, 4.5, 5.0, 5.5, and 6%. The plots for the stability, flow, density, and volumetric properties (AV% and VMA%) are presented in Figure 8. The optimum asphalt content (OAC) was found to be 4.9 percent. At this content, the flow value as well as the VMA% were within the specification limit: 2 to 4 mm for the former and greater than 14% for the latter. The OAC of 4.9% was kept constant for other mix types instead of optimizing the mix design for each mix type since the primary goal of this research was to investigate the ability of asphalt concrete overlays to mitigate reflection cracking. 

## 4. OT Results and Discussion

The results of the OT test are presented in the following articles in terms of variable overlay thicknesses (40 mm, 50 mm, and 60 mm) and temperatures (20 °C, 30 °C, and 40 °C) for each type of specimen (i.e., reference mix without any reinforcement type, overlay specimen reinforced by geotextiles located at the bottom, overlay specimen located at one-third of the depth from the bottom, and mixes modified with chopped fibers). In order to best characterize the reflection cracking during the crack initiation phase and propagation phase, four parameters were considered for discussion: *Gc*, CPR, maximum tensile load, and Nf. 

### 4.1. Reference Specimens

The performance of the reference specimens (without any reinforcement type) under varying overlay thicknesses and temperature conditions was evaluated based on four parameters—*Gc*, CPR, maximum tensile load at the first load cycle, and Nf—and the results are shown graphically in Figure 9. It is evident that as the overlay thickness increased from 40 mm to 60 mm there was a consistent improvement in all four parameters across all temperature levels (20, 30, and 40 °C). For example, at a temperature of 20 °C, the critical fracture energy (*Gc*) increased from 0.468 (for 40 mm) to 0.630 (for 60 mm). This showed a performance improvement rate of about 34.5%. Similarly, the number of load cycles to failure (Nf) improved from 224 to 344, suggesting a 53.6% increase in the number of load cycles the overlay could withstand before failure. This clearly suggests that an increase in overlay thickness enhances the performance by improving the overlay’s crack resistance as well as better retarding the initiation of the crack, as indicated by the results of the maximum tensile load. On the other hand, with the increase in temperature, the performance of the reference specimens showed a consistent decline across all thickness levels (40 mm, 50 mm, and 60 mm). For instance, considering the 40 mm overlay, the critical fracture energy (*Gc*) dropped from 0.468 (at 20 °C) to 0.236 (at 40 °C), showing a performance degradation of about 49.5%. Similarly, the number of load cycles to failure (Nf) declined from 224 to 78, showing a 65.2% decrease in the number of cycles it could endure before failure. Additionally, in comparison to the specimens tested at 20 °C, the maximum tensile load to initiate cracking in the overlay showed reduction rates of 14.12% and 28.37% for specimens tested at 30 °C and 20 °C, respectively. This reduction in performance parameters indicates that higher temperatures can weaken the asphalt material, making it more susceptible to crack initiation and propagation. In conclusion, the data suggest that increasing the thickness of the asphalt overlay can significantly enhance its performance by improving its resistance to crack initiation and propagation. Conversely, high temperatures can deteriorate performance by accelerating crack onset and propagation. Thus, in warmer climates, the asphalt overlay thickness might need to be increased to compensate for the reduction in performance caused by higher temperatures. These findings emphasize the importance of considering local temperature conditions in the design of asphalt overlays.

### 4.2. Geotextiles at the Bottom

Figure 10 depicts the performance of specimens with geotextiles placed at the bottom in terms of *Gc*, CPR, maximum tensile strain, and Nf under varying overlay thicknesses and temperature conditions. All temperature levels (20, 30, and 40 °C) exhibited a similar pattern of enhanced performance as the thickness of the overlay increased from 40 mm to 60 mm. At 20 °C, for instance, the critical fracture energy (*Gc*) increased from 0.586 (for 40 mm thickness) to 0.867 (for 60 mm thickness), representing an enhancement rate of approximately 47.8%. The number of load cycles to failure (Nf) improved from 394 to 521, indicating a 32.2% increase. The above results are in agreement with those of Khodaei and Falah [19]. In addition, as the thickness increased, the maximum tensile load to initiate reflection cracking increased. For example, at 20 °C, the tensile load increased by approximately 60% (from 2558 N to 4087 N) as the thickness rose from 40 mm to 60 mm. Based on the temperature effect on reflection cracking parameters, as the temperature increased, there was a decline in performance across all thickness levels (40 mm, 50 mm, and 60 mm). For the 40 mm overlay, the critical fracture energy (*Gc*) dropped from 0.586 (at 20 °C) to 0.307 (at 40 °C), representing a performance reduction of about 47.6%. Additionally, the CPR increased by 40% when the temperature rose from 20 °C to 40 °C. Similarly, the maximum tensile load and the number of load cycles to failure (Nf) dropped from 2558 N to 1715 N and 394 to 98, respectively, representing a reduction of about 49% and 75.1%. These results indicate that higher temperatures can weaken the asphalt concrete and make the geotextile less stiff, making the specimen more prone to cracking. However, when compared to the reference specimens, the overlays with geotextiles at the bottom showed a better performance at all thickness and temperature levels. For instance, at 20 °C and 40 mm thickness, the reference mix had a *Gc* of 0.468, while the mix with geotextiles at the bottom had a *Gc* of 0.586, indicating an improvement of about 25.2%. The enhanced performance with geotextiles at the bottom can be attributed to the added reinforcement that the geotextile provides, which likely improves the crack initiation resistance and crack propagation as well. In conclusion, the findings reveal that the use of geotextile at the bottom of the overlay significantly enhances the performance compared to the reference specimen due to its reinforcement properties. These insights can be beneficial in the design and construction of asphalt overlays, especially in areas with varying temperature conditions.

### 4.3. Geotextiles at One-Third of the Depth

The OT results for specimens with geotextiles positioned at one-third of the depth from the bottom are exhibited in Figure 11. Similar to the previous cases, the performance improved with an increase in overlay thickness from 40 mm to 60 mm across all temperature levels (20, 30, and 40 °C). For instance, at a temperature of 20 °C, the critical fracture energy (*Gc*) increased from 0.653 (for 40 mm thickness) to a remarkable 1.184 (for 60 mm thickness), indicating an improvement rate of about 81.3%. Additionally, there was a reduction in the CPR and an enhancement in the maximum tensile load. A noticeable decrease in the CPR, around 27%, was recorded when the overlay thickness increased from 40 mm to 60 mm at 20 °C. This indicates a slower rate of crack propagation in overlays with greater thicknesses. Concurrently, the maximum tensile load experienced an improvement of about 73% under the same conditions, suggesting the increased ability of thicker overlays to withstand crack initiation. Similarly, the number of load cycles to failure (Nf) also improved from 688 to 817, showing an improvement rate of approximately 18.8%.

Regarding the temperature influence on reflection cracking parameters, it is evident that as the temperature increased, there was an expected decline in performance across all thickness levels (40 mm, 50 mm, and 60 mm). For the 40 mm overlay, the critical fracture energy (*Gc*) dropped from 0.653 (at 20 °C) to 0.370 (at 40 °C), demonstrating a performance reduction of about 43.3%. Additionally, this rise in temperature led to an increase in the CPR and a decrease in the maximum tensile load by 100% for the former and 31% for the latter, pointing to an accelerated crack propagation rate with higher temperatures. Similarly, the number of load cycles to failure (Nf) dropped from 688 to 184, signifying a reduction of about 73.3%. These results imply that higher temperatures may lead to increased susceptibility to cracking. 

However, when compared with the reference specimens, the overlays with geotextiles at one-third of the depth from the bottom showed significantly better performance across all thickness and temperature levels. For instance, the *Gc* value at 20 °C and 60 mm thickness in the specimen with geotextiles at one-third depth was 1.184, while it was 0.630 in the reference specimen with 60 mm thickness, a relative improvement of around 88%; in addition, the corresponding decrease in the CPR was about 38%. The Nf for the specimen with geotextiles was 817, compared to 344 for the reference specimen, an improvement of nearly 137%. The above results collectively denote a significant enhancement in the performance of the overlay regarding crack initiation resistance as well as propagation when geotextiles are positioned at one-third of the depth from the bottom in asphalt overlays in comparison to the reference specimen.

### 4.4. Chopped Fibers

The results of the specimens with chopped fibers are presented in Figure 12. The pattern of performance improvement with an increase in overlay thickness from 40 mm to 60 mm was consistent across all temperature levels (20, 30, and 40 °C), similar to those in previous cases, but to a different extent. For each increment in thickness of 10 mm, the *Gc*, maximum tensile stress, and Nf increased by a rate of 16.6%, 26.2%, and 29.2%, respectively, whereas the CPR decreased at a rate of 11.1%. These results also denote the high resistance to the onset of reflection cracking and propagation. However, with the increase in temperature, the performance across all thickness levels (40, 50, and 60 mm) declined. For the 40 mm overlay, the critical fracture energy (*Gc*) reduced from 0.483 (at 20 °C) to 0.238 (at 40 °C), denoting a performance reduction of about 50.7%. Similarly, the number of load cycles to failure (Nf) dropped from 242 to 81, indicating a reduction of approximately 66.5%. This suggests that the benefits of incorporating chopped fibers into the overlay may be offset at higher temperatures due to the increased susceptibility to cracking. However, when comparing the performance of overlays with chopped fibers against the reference specimens, there was a slight improvement across all thickness and temperature levels. At 20 °C and 40 mm thickness, the reference mix had a *Gc* of 0.468, while the mix with chopped fibers achieved a *Gc* of 0.483, marking an improvement of around 3.2%. On the other hand, the average CPR value over the investigated thicknesses was approximately similar to those of the reference mix.

### 4.5. Comparison of Reinforced Specimens Performance

In order to compare the results of the OT for specimens reinforced with geotextiles at different positions and chopped fibers with those of reference specimen, the average value was calculated for each parameter of reflection cracking, i.e., *Gc*, CPR, maximum tensile load at the first loading cycle and Nf, based on thickness and temperature to more clearly show the effect of reinforcement type on each parameter. As shown in Figure 13, it’s obvious that the overlays with geotextiles, whether at the bottom or one-third of the depth from the bottom, have higher average *Gc* values compared to the reference specimen and overlay with chopped fibers. This indicates that they possess higher resistance to fracture and they can withstand more stress before initiating cracking. Interestingly, the overlay with geotextiles at one-third of the depth from the bottom exhibited the highest average *Gc*. This could be attributed to the ability of the geotextile layer to strengthen the middle portion of the overlay, which is generally subjected to the most tensile stress, thereby enhancing its fracture resistance. The improvement rate in the *Gc* parameter for the specimen with geotextiles at one-third of the depth from the bottom was 66.1%, 20.3%, and 63.1% in comparison with the reference and the specimens with geotextiles at the bottom and chopped fibers, respectively. Based on the of Sadek et al. (2020), the acceptable limit of *Gc* is 0.5, which differentiates between a soft mix (lower than 0.5 N.mm/ mm^2^) and tough mix (equal or higher than 0.5 N.mm/mm^2^). The tough mixes for asphalt concrete overlay provide high resistance to the initiation of reflection cracking. Reviewing the results of *Gc* exhibited in Figure 13 showed that the specimens reinforced with geotextiles (either at the bottom or at one-third of the depth from the bottom) demonstrated acceptable resistance to reflection crack initiation.

Meanwhile, the results of the CPR, or crack propagation rate, which gives insight into the rate at which a crack progresses through a material under stress, showed that the overlay with chopped fibers and the reference mix had relatively high CPR values, suggesting a faster crack propagation rate. However, the overlays with geotextiles, especially at one-third of the depth from the bottom, had relatively low CPR values, meaning they exhibited slower crack propagation. This could be due to the reinforcing effect of the geotextiles, which could potentially impede the growth of cracks and thus contribute to prolonging the lifespan of the asphalt overlay. In comparison to the reference specimen, there were reductions of about 39.4%, 11.4%, and 4.3% when specimens were reinforced with geotextiles at one-third of the depth from the bottom, geotextiles at the bottom, and chopped fibers, respectively. Sadek et al. [29] stated that the CPR limit of 0.5 can be used to differentiate between the crack-resistant specimen (lower than or equal to 0.5) and crack-susceptible specimen (higher than 0.5). Based on these criteria, only mixes reinforced with geotextiles can pass the suggested limit and provide specimens with good resistance to crack propagation. 

When observing the effect of geotextile positioning on the maximum tensile load, a distinct trend emerges. The overlay with geotextiles at the bottom presented an average improvement of around 7.3% in the maximum tensile load, demonstrating its remarkable ability to withstand tension at the first load cycle. However, the geotextile’s influence became even more prominent when placed at one-third of the depth from the bottom. Under this arrangement, the overlay showed an impressive improvement rate of 26.4% in maximum tensile load. The results clearly signify that the geotextile’s position within the overlay significantly affects its load-resistance capacity, with the one-third depth arrangement offering the highest advantage. However, using chopped fibers to reinforce the asphalt concrete mixes did not have a significant effect on the maximum tensile load compared to the reference specimen.

Turning to the results of Nf, or the number of load cycles to failure, the geotextile’s positive impact is clearly shown. The average improvement rate with geotextiles at the bottom was around 44.6% in comparison to the reference specimen, suggesting enhanced durability against reflection cracking. However, the benefits were raised remarkably when the geotextile was positioned at one-third of the depth, exhibiting an almost doubled (97.8%) improvement rate. This finding emphasizes the geotextile’s crucial role in augmenting the lifespan of overlays. In contrast to the above findings, the implementation of chopped fibers did not seem to offer any improvement in the asphalt concrete lifespan based on the performance against reflection cracking. 

## 5. Conclusions and Recommendations

The main objective of this research was to investigate the impact of asphalt concrete overlay reinforcement using geotextiles and chopped fibers under varying thicknesses (40, 50, and 60 mm) and temperatures (20, 30, and 40 °C). This comprehensive analysis inspected different performance parameters, such as the critical fracture energy (*Gc*), crack propagation rate (CPR), maximum tensile load, and the number of load cycles to failure (Nf). Four types of specimens were examined: the reference mix and three modified asphalt concrete overlays, one with geotextiles at the bottom, one with geotextiles placed at one-third of the depth from the bottom, and one modified with chopped fibers. Based on the test results obtained in this study, the following conclusions can be drawn:The presence of geotextiles, whether at the bottom or one-third of the depth from the bottom, significantly improved the overall performance of the asphalt concrete overlays. For instance, overlays with geotextiles at one-third of the depth from the bottom demonstrated an average improvement in *Gc* and Nf of approximately 66.1% and 97.8%, respectively, compared to the reference mix.As the overlay thickness increased from 40 mm to 60 mm, there were noticeable improvements in the performance against reflection cracking for all parameters across all specimens. For example, in the reference mix, *Gc* values increased by around 34.5% when the overlay thickness changed from 40 mm to 60 mm, and the number of load cycles to failure (Nf) improved from 224 to 344, suggesting a 53.6% increase in the number of load cycles the overlay could withstand before failure.The performance parameters *Gc*, CPR, maximum tensile load, and Nf exhibited dependence on the test temperature, which became more pronounced in the modified overlay types. For instance, the reference mix at 20 °C experienced around 61% more load cycles to failure (Nf) than at 40 °C, whereas the corresponding rate was 67% in the case of specimens with geotextiles at the bottom.Overlays with geotextiles at one-third of the depth from the bottom consistently outperformed the others in terms of *Gc*, CPR, maximum tensile load, and Nf, indicating their superior resistance to crack initiation and propagation.Despite a slight enhancement in certain properties, such as the *Gc* value at 20 °C and 40 mm thickness, the incorporation of chopped fibers in the overlays did not substantially improve the overall performance in comparison to the reference specimens.Based on the above results, which are limited to the available materials and testing programs, it is suggested to consider different types of asphalt cement as well as aggregate gradations. Additionally, extending the temperature range both below and above those examined here will more accurately reflect the climatic conditions of both colder and hotter regions.

## Figures and Tables

**Figure 1 materials-16-05990-f001:**
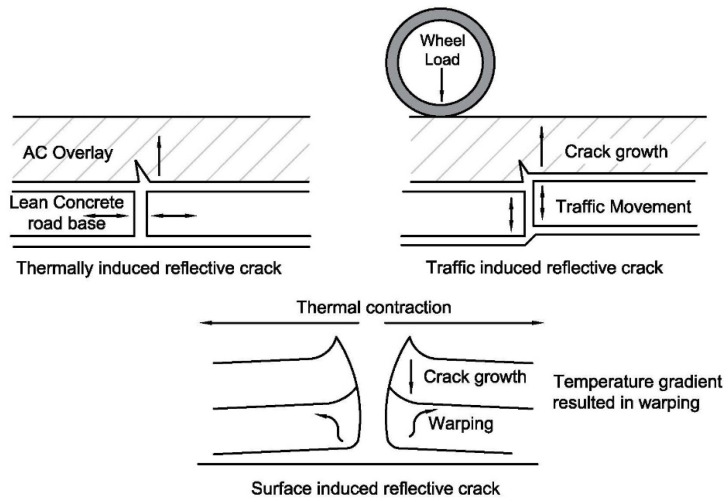
Mechanisms of reflection cracking.

**Figure 2 materials-16-05990-f002:**
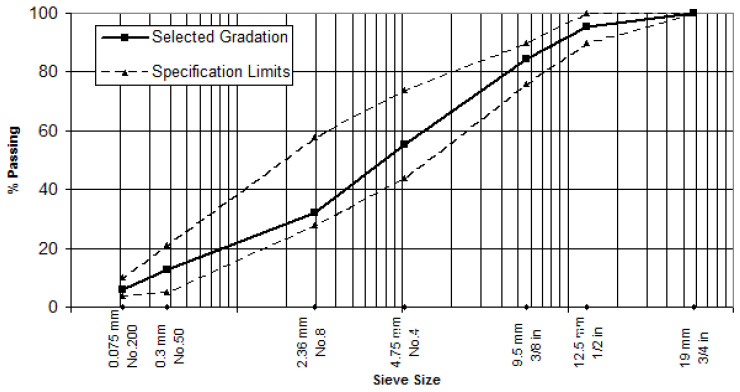
Gradation of aggregate.

**Figure 3 materials-16-05990-f003:**
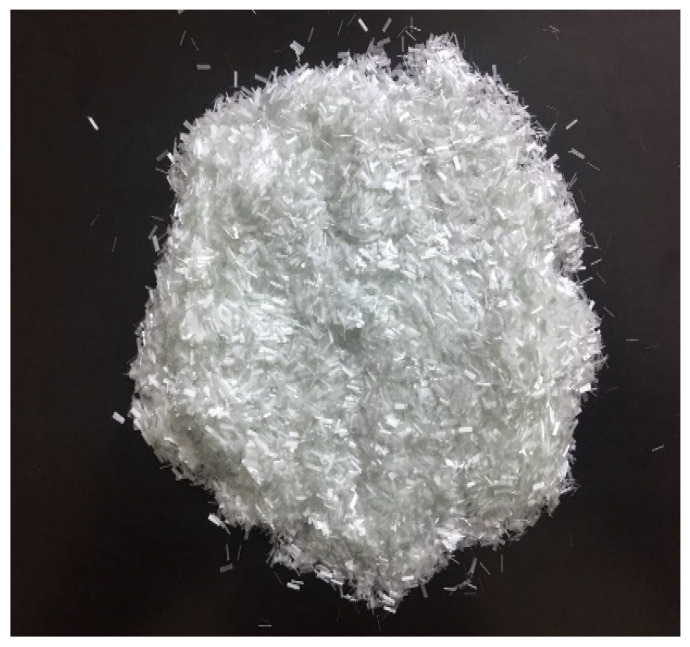
Chopped Glass Fibers.

**Figure 4 materials-16-05990-f004:**
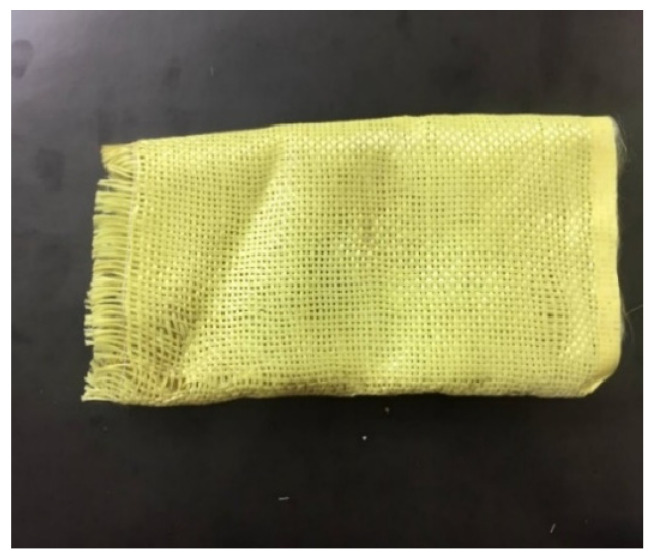
Kevlar fiber geotextile.

**Figure 5 materials-16-05990-f005:**
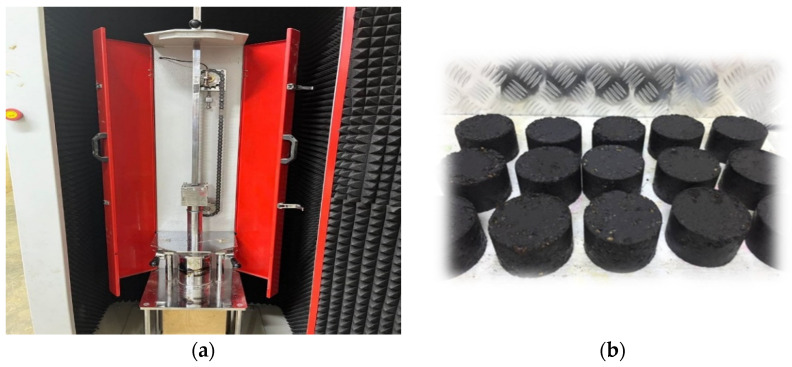
Compaction methods and specimens. (**a**) Marshall compactor; (**b**) Marshall Specimens; (**c**) Slab specimen; (**d**) Roller compactor.

**Figure 6 materials-16-05990-f006:**
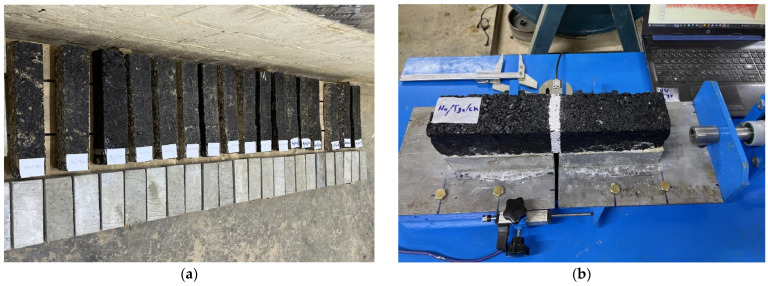
Specimens and AC specimens being tested. (**a**) PCC and AC specimens; (**b**) test setup.

**Figure 7 materials-16-05990-f007:**
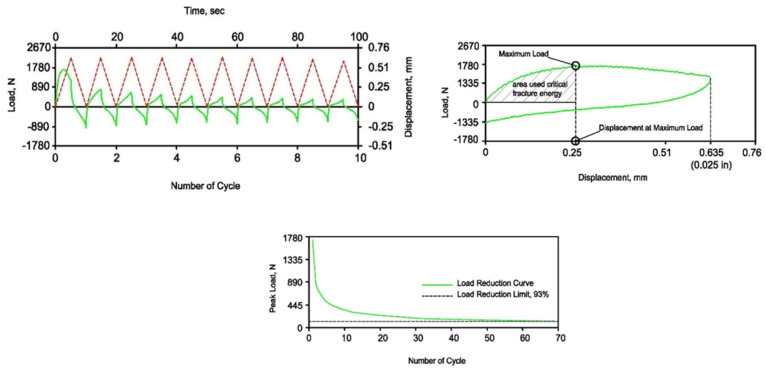
Typical OT outputs.

**Figure 8 materials-16-05990-f008:**
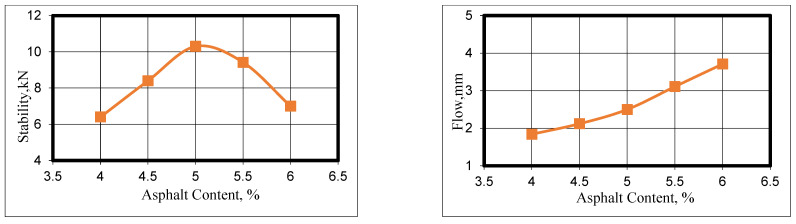
Mix design results via the Marshall method.

**Figure 9 materials-16-05990-f009:**
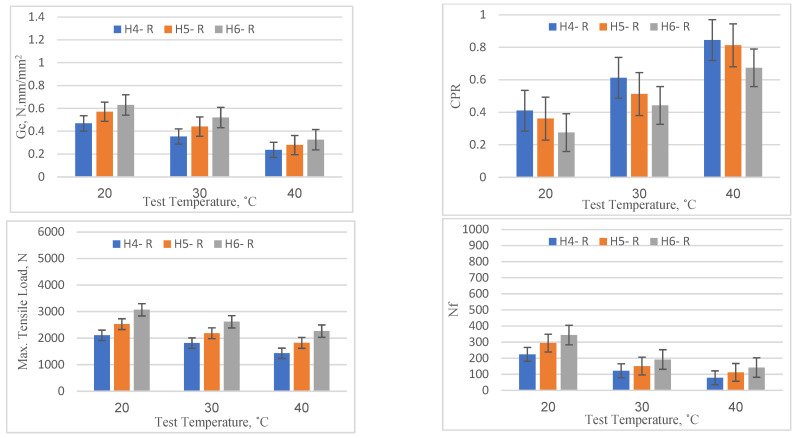
Reflection cracking parameters for reference specimen.

**Figure 10 materials-16-05990-f010:**
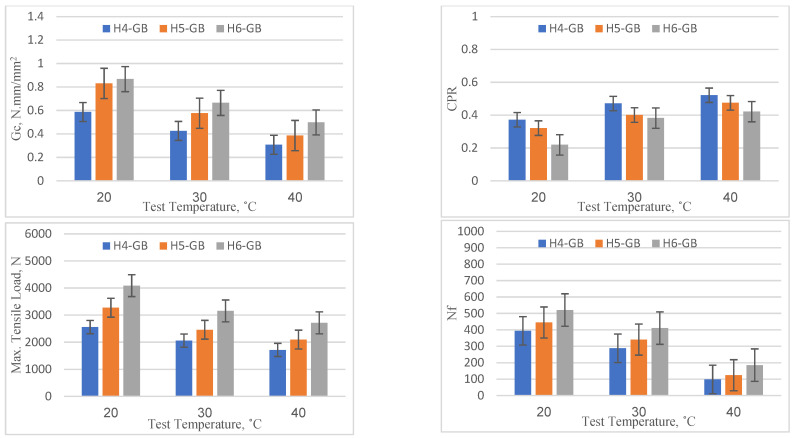
Reflection cracking parameters for specimen with geotextiles at the bottom.

**Figure 11 materials-16-05990-f011:**
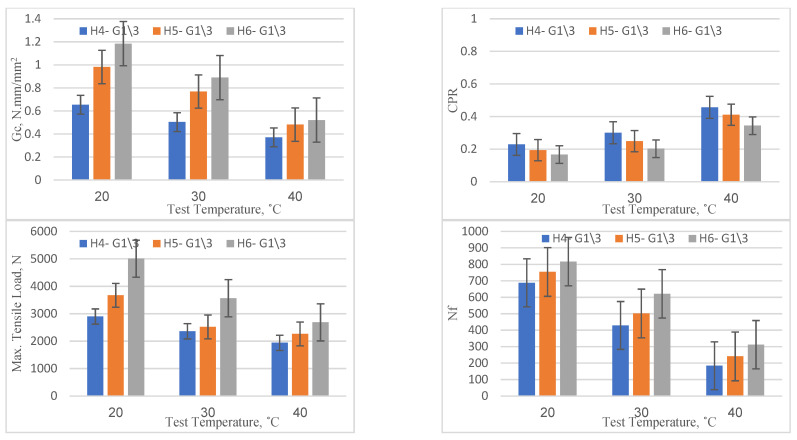
Reflection cracking parameters for specimen with geotextiles at one-third of the depth from the bottom.

**Figure 12 materials-16-05990-f012:**
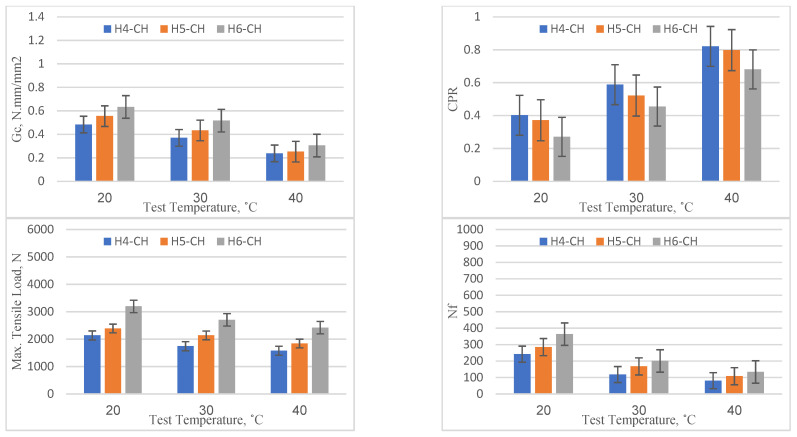
Reflection cracking parameters for specimen with chopped fibers.

**Figure 13 materials-16-05990-f013:**
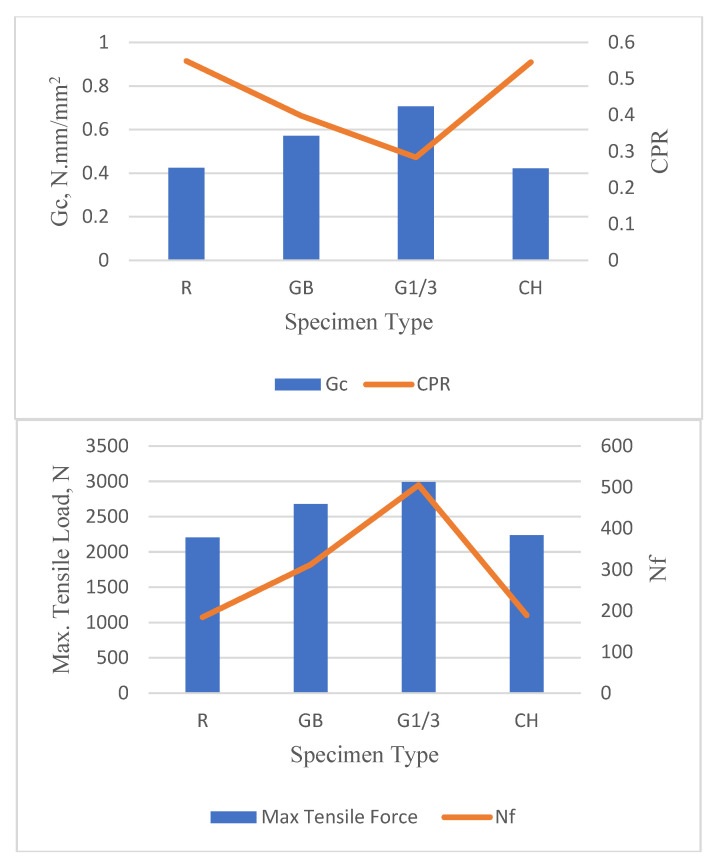
Comparison of reflection cracking parameters for different specimens.

**Table 1 materials-16-05990-t001:** A review of the reflection cracking mitigation and testing methods.

Method	Researcher	Description	Test	Effect on Reflection Cracking
Modifying asphalt concrete	[11,12]	Carbon black (CB). Low-density polyethylene LDPE). Styrene butadiene styrene (SBS). LDPE and SBS. LDPE and CB.	Dynamic cyclic loading test (flexural mode)	The best improvement in the number of loading cycles to initiate cracking was obtained using LDPE and CB
	[13]	Polymer-modified asphalt (PMA) with usual dosage (3 and 3.2% SBS). PMA with high dosage (7.6 and 8% SBS).	Texas overlay test	The high content of PMA resulted in better reflection cracking resistance than PMA mixes with normal content
	[14]	Natural zeolite was added to crumb-rubber-modified (CRM) asphalt binder. Hydrated lime was added to the CRM asphalt binder.	Beam fatigue test Dynamic cyclic loading test	CRM binders with natural zeolite are superior in crack mitigation in comparison to CRM binders with hydrated lime
Incorporation of fibers	[15]	Asphalt concrete modified with polypropylene fibers.	Field study	Crack severity was less on the fiber-modified overlay sections
	[16]	Asphalt concrete modified with 0.3%, 0.5%, and 1% polypropylene fibers.	Repeated indirect tension test	Less reflection cracking exhibited for modified mixes in comparison to the reference mix
The use of geosynthetic stress-relieving layer	[17]	Geogrid type Tensar Biaxial (BX 1500), Polypropylene fabric.	Dynamic cyclic loading test.	The rate of crack propagation was remarkably reduced compared to unreinforced sample.
	[18]	Fiberglass grid.	Three-point cyclic bending test.	The reinforcement with the grid system resulted in better resistance to crack propagation.
	[19]	Polyester geogrid.	Dynamic cyclic loading test on the rubber base.	Crack growth was drastically slowed when a geogrid was installed at a depth of one-third of the overlay.
	[20]	Fiberglass geogrid. Polyester geogrid.	Cyclic loading test for specimen placed on the rubber base.	In a reinforced overlay, a glass grid is superior to a polyester grid
	[21]	Non-woven geotextile.	Texas overlay test.	When used as an interlayer system, geotextiles delay the spread of reflection cracks.
Thick overlay thickness	[22]	Two thicknesses of overlay (75 and 100 mm).	Cyclic loading test for specimen placed on the rubber base.	A decrease in asphaltic concrete thickness of 25% produced a corresponding decrease in reflection cracking life by 400%.
	[20]	Three thicknesses of overlay (50, 70, and 90 mm).	Cyclic loading test for specimen placed on the rubber base.	A thicker overlayer offered better resistance to reflection cracking.
Geosynthetic Inclusion of fiber Thick overlay thickness	Current Study	Two different locations: bottom and one-third depth from the bottom. 10% chopped glass fiber by weight of asphalt cement. Three thicknesses (40, 50, and 60 mm).	Texas overlay test.	….

**Table 2 materials-16-05990-t002:** Performance Grade Properties of Asphalt Cement.

Asphalt Cement	Properties	Temperature Measured (°C)	Measured Parameters	Specification Requirements, AASHTO M320-05
Original	Flash point (°C)	-	284	230 °C, min
	Viscosity at 135 °C (Pa.s)	-	0.466	3 Pa.s, max
	DSR, G/sinδ at 10 rad/s (kPa)	58	3.3044	1.00 kPa, min
		64	2.2182	
		70	0.912	
RTFO Aged	Mass loss (%)	-	0.621	1%, max
	DSR, G/sinδ at 10 rad/s (kPa)	58	4.1669	2.2 kPa, min
		64	3.1287	
		70	1.9118	
PAV Aged	DSR, G/sinδ at 10 rad/s (kPa)	28	4499	5000 kPa, max
		25	6247	
	BBR, creep stiffness (MPa)	−6	123.7	300 MPa, max

**Table 3 materials-16-05990-t003:** Aggregate Physical Properties.

Test	ASTM Standard	Result	Specification of SCRB/R9 [23]
Coarse Aggregate			
Apparent specific gravity	C127	2.641	-
Bulk specific gravity		2.628	-
Water absorption (%)		0.208	-
Soundness (sodium sulfate solution loss; %)	C88	2.7	12 max.
Percent wear (Los Angeles abrasion; %)	C131	21	30 max.
Flat and elongated (5:1; %)	D4791	4	10 max.
Fractured Pieces (%)	D5821	96	90 min.
Fine Aggregate			
Apparent specific gravity Bulk specific gravity Water absorption (%)	C128	2.608 2.542 0.872	- - -
Clay lump and friable particles (%)	C142	1.08	3 max.
Sand equivalent (%)	D2419	51	45 min.

**Table 4 materials-16-05990-t004:** Mineral Filler Physical Properties.

Test	Results
Specific gravity	3.15
Passing sieve No. 200 (%)	98

**Table 5 materials-16-05990-t005:** Properties of the Chopped Fibers.

Trade code	ECS13-06-562A
Type of glass	E
Filament diameter (µm)	13
Chop length (mm)	12
Tensile strength (MPa)	1340
Density (kg/m^3^)	1903

**Table 6 materials-16-05990-t006:** Properties of the Kevlar fiber geotextile.

Product type	Woven aramid fabric
Style/pattern	Twill/unidirectional
Material	poly-phenylene terephthalamide
Thickness	1 mm
Color	Yellow
Tensile strength (gpd)	23
Density (kg/m^3^)	1446

**Table 7 materials-16-05990-t007:** The Nomenclature of Asphalt Concrete Specimens.

No.	Nomenclature	Height (mm)	Reinforcement	
			Type	Location
1	H4-R	40	Nil	…
2	H5-R	50	Nil	…
3	H6-R	60	Nil	...
4	H4-GB	40	Geotextile	Bottom
5	H5-GB	50	Geotextile	Bottom
6	H6-GB	60	Geotextile	Bottom
7	H4-G1\3	40	Geotextile	1\3 depth from the bottom
8	H5-G1\3	50	Geotextile	1\3 depth from the bottom
9	H6-G1\3	60	Geotextile	1\3 depth from the bottom
10	H4-CH	40	Chopped	Added to mix
11	H5-CH	40	Chopped	Added to mix
12	H6-CH	40	Chopped	Added to mix

## Data Availability

The data presented in this study are available on request from the corresponding author.

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
