# Peer review of "Comparative Analysis of Reinforced Asphalt Concrete Overlays: Effects of Thickness and Temperature"

_materials, 2023, doi:10.3390/ma16175990_

Round 1

Reviewer 1 Report

Attached separately

Author Response

The authors wish to thank the reviewer for the time and effort in reviewing our manuscript. The provided comments and constructive suggestions are very helpful in improving the quality of our manuscript and have helped us identify several shortcomings in our manuscript. All the provided comments were addressed.

Reviewer 2 Report

This paper investigates the anti-reflective cracking performance of asphalt concrete over-lays, based on the change of thickness and temperature of asphalt concrete over-lays, as well as the incorporation of fibers and geotextiles, to compare and analyze the anti-reflective cracking performance of asphalt paving layers under different conditions. Good idea and experimental design, the paper would be better if some problems in the text were solved. The details are raised as follows:

1) Figures: The fonts in the figures should be formatted in Times New Roman.

2) Tables: Use of three-line tables in the paper.

3) Table 2: The precision should be guaranteed to be the same for the same type of data.

4) The way chopped fiber is incorporated into the asphalt mix needs to be described in the paper.

5) Briefly describe, how the geotextile is added to a depth of 1/3 from the bottom of the specimen.

6) Line 168 and Line 172: “Some photographs presented in Figure 4 show...” should be “Some photographs presented in Figure 5 show...”; “Table 5 listed the nomenclature...” should be “Table 7 listed the nomenclature...”. Check the entire paper to determine the correct figure and table serial number correspondences.

7) 4.4. Chopped fiber: Explain the reason why chopped fiber incorporated does not improve parameters such as Gc.

8) Line 435: “... the following conclusions can be drawn;” should be “...the following conclusions can be drawn:”

Considering above-mentioned reasons, I suggest this paper needs a minor revision.

Minor editing of English language required.

Author Response

(The authors gave the same response as above.)

Reviewer 3 Report

The paper examines how the properties of reinforced asphalt concrete overlays are influenced by their thickness and temperature. This concept is both original and relevant, addressing a crucial topic, especially given the escalating concerns about the detrimental effects of global warming on infrastructure.

There are several intriguing facets associated with this manuscript. The primary significance of this paper lies in its investigation of the interactive effects of two pivotal factors on concrete properties: temperature and thickness.

It's worth noting that Figure 1 appears to contain copyrighted material. If this is indeed the case, the authors might need to secure permission to use it.

Furthermore, there are instances where the journal's formatting requirements for numbering headings and subheadings haven't been fully adhered to. For instance, in line 103, 'Asphalt cement' is a subheading that should be numbered, as per my understanding.

The abstract still has room for improvement. I suggest incorporating some of the key quantitative results from the study.

Including practical insights and recommendations for practitioners would be valuable. This would shed light on the real-world implications of the study's findings.

Expanding the conclusion section is advisable. One way to achieve this is by providing recommendations for future research directions.

While the temperature range considered in this study might not be extensive, it would be interesting if the authors could touch on a less-known phenomenon observed in some concrete members exposed to temperature called 'fire-induced spalling of concrete.' This could be based on their experimental observations.  

The references provided are sufficient.

Author Response

(The authors gave the same response as above.)

Round 2

Reviewer 1 Report

The manuscript now can be accepted for publication.